# You Only Debias Once: Towards Flexible Accuracy-Fairness Trade-offs at Inference Time

Xiaotian Han[1], Tianlong Chen[2], Kaixiong Zhou[3], Zhimeng Jiang[4], Zhangyang Wang[5], Xia Hu[6]
[1]Case Western Reserve University, [2]University of North Carolina at Chapel Hill, [3]North Carolina State University, [4]Texas A&M University, [5]University of Texas at Austin, [6]Rice University
xhan@case.edu, tianlong@cs.unc.edu, kzhou22@ncsu.edu, zhimengj@tamu.edu,
atlaswang@utexas.edu, xia.hu@rice.edu

Deep neural networks are prone to various bias issues, jeopardizing their applications for high-stake decision-making. Existing fairness methods typically offer a fixed accuracy-fairness trade-off, since the weight of the well-trained model is a fixed point (fairness-optimum) in the weight space [1]. Nevertheless, more flexible accuracy-fairness trade-offs at inference time are practically desired since: 1) stakes of the same downstream task can vary for different individuals, and 2) different regions have diverse laws or regularization for fairness. If using the previous fairness methods, we have to train multiple models, each offering a specific level of accuracy-fairness trade-off. This is often computationally expensive, time-consuming, and difficult to deploy, making it less practical for real-world applications. To address this problem, we propose *You Only Debias Once* (YODO) to achieve in-situ flexible accuracy-fairness trade-offs at inference time, using *a single model* that trained only once. Instead of pursuing one individual fixed point (fairness-optimum) in the weight space, we aim to find a "line" in the weight space that connects the accuracy-optimum and fairness-optimum points using a single model. Points (models) on this line implement varying levels of accuracy-fairness trade-offs. At inference time, by manually selecting the specific position of the learned "line", our proposed method can achieve arbitrary accuracy-fairness trade-offs for different end-users and scenarios. Experimental results on tabular and image datasets show that YODO achieves flexible trade-offs between model accuracy and fairness, at ultra-low overheads. For example, if we need 100 levels of trade-off on the ACS-E dataset, YODO takes 3.53 seconds while training 100 fixed models consumes 425 seconds. The code is available at https://github.com/ahxt/yodo.

## 1. Introduction

Deep neural networks (DNNs) are prone to bias with respect to sensitive attributes [1–6], raising concerns about their application on high-stake decision-making, such as credit scoring [7], criminal justice [8], job market [9], healthcare [10–13] and education [14–16]. Decisions made by these biased algorithms could have a long-lasting, even life-long impact on people's lives and may affect underprivileged groups negatively. Existing studies have shown that achieving fairness involves a trade-off with model accuracy [17–20]. Therefore, various methods are proposed to address fairness while maintaining accuracy [21–23], but typically, these methods have been designed to achieve a fixed level of the accuracy-fairness trade-off. In real-world applications, however, the appropriate trade-off between accuracy and fairness may vary depending on the context and the needs of different stakeholders/regions. Thus, it is important to have flexible trade-offs at inference time due to:

---

[1]The *weight space* of a model refers to the space of all possible values that the model's trainable parameters can take. If a neural network has $m$ trainable parameters, then the dimension of the weight space is $m$. Each point, a $m$-dimensional vector, in the weight space corresponds to a specific model. For visualization purposes, the weight space is often reduced to a 2D space, as shown in Figure 1.

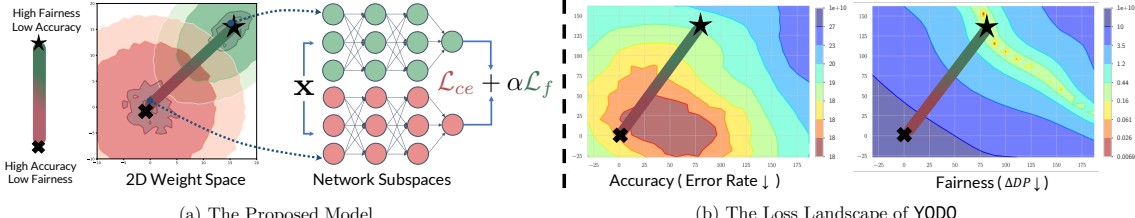

(a) The Proposed Model  (b) The Loss Landscape of YODO

Figure 1: (**a**):The overview of our proposed method. 2D Weight Space indicated the landscape of model accuracy and fairness. ✖ indicates the accuracy-optimum weight with high accuracy but low fairness, and ★ indicates the fairness-optimum weight with low accuracy but high fairness. Network Subspaces shows the different subspaces correspond with different objectives (i.e., accuracy $\mathcal{L}_{ce}$ and fairness $\mathcal{L}_f$). (**b**): The loss landscape of the model accuracy (error rate) and fairness ($\Delta$DP) in the same weight space of our proposed method. The weight space is reduced to two dimensions [27]. The different points indicate different objectives, ✖ indicates the accuracy-optimum endpoint in the weight space, while ★ indicates the fairness-optimum endpoint in the weight space. The dataset is ACS-I with gender as the sensitive attribute. $\Delta$DP is the demographic parity difference, which asserts that the probability of a positive outcome should be the same across all demographic groups.

**1) Downstream tasks with different stakes can have varying fairness requirements, depending on the individuals involved.** According to a survey by Srivastava et al. [24], people prioritize accuracy over fairness when the stakes are high in certain domains. For example, in healthcare (e.g., cancer prediction), accuracy should be favored over fairness, as prioritizing fairness can lead to "fatal" consequences, such as missing cancer diagnoses at higher rates [24, 25]. Thus, in high-stakes domains, accuracy-fairness trade-off should be flexible and controllable at inference time.
**2) Different regions have different laws or regulations for fairness** The use of decision-making systems is regulated by local laws and policies. However, countries may exhibit differences in the importance of fairness in various applications. For example, the labor market may expect fairness to be much stronger in Germany than in North America [26]. Therefore, developers should consider the varying fairness requirements when applying their algorithms in different regions.

Unfortunately, while urgently needed, flexible accuracy-fairness trade-offs at inference time remain underexplored. Thus, we aim to answer the following question in this paper: *Can we achieve flexible accuracy-fairness trade-offs at inference time using a single model that is trained only once?*

It is an open but challenging problem to achieve flexible accuracy-fairness trade-offs. One solution is to train multiple models for different trade-offs. However, this is less practical due to the significant training time and memory overhead. Another solution is the post-processing fairness method, which may lead to suboptimal model accuracy and require sensitive attributes at inference time [28]. To address the above question, we propose *You Only Debias Once* (YODO) to achieve flexible fairness-accuracy trade-offs via learning an *objective-diverse* neural network subspace that contains accuracy-optimum and fairness-optimum points in the weight space. As illustrated in Figure 1(a), we design an objective-diverse neural network subspace, which contains two endpoints (the red network and the green network in the weight space). During training time, the two endpoints are encouraged to accuracy-optimum (✖ in Figure 1) and fairness-optimum (★ in Figure 1) solutions, respectively. The "line" between the two endpoints is encouraged to achieve transitional solutions. At inference time, we achieve arbitrary accuracy-fairness trade-offs by manually selecting a trade-off coefficient to determine the model weight (i.e., select a position of the line). Our **contributions** are as follows:

- We propose a novel approach that allows for flexible accuracy-fairness trade-offs during inference, despite being trained only once. This adaptability aligns with the varying fairness demands of real-world applications.
- We achieve the above in-situ flexible accuracy-fairness trade-offs by introducing an **objective-diverse** neural network subspace. The subspace has two different endpoints in weight space, which are optimized for accuracy-optimum and fairness-optimum and the "line" between the two endpoints to achieve transitional solutions. Thus, it can achieve flexible trade-offs by customizing the endpoints at inference time, with ultra-low overheads.
- Experimental results validate the effectiveness and efficiency of the proposed method on both tabular and image data. The result of experiments shows that our proposal achieves comparable

performance with only one training when compared to trained models for a single level of accuracy-fairness trade-off.

## 2. Preliminaries

In this section, we introduce the notation used throughout this paper and provide an overview of the preliminaries of our work, including algorithmic fairness and neural network subspace learning.

**Notations.** We use $\{\mathbf{x}, y, s\}$ to denote a data instance, where $\mathbf{x} \in \mathbb{R}^d$, $y \in \{0, 1\}$, $s \in \{0, 1\}$ are feature, label, sensitive attribute, respectively. $\hat{y} \in [0, 1]$ denotes the predicted value by machine learning model $\hat{y} = f(\mathbf{x}; \theta) : \mathbb{R}^d \to [0, 1]$ with trainable parameter $\theta \in \mathbb{R}^m$ in the $m$-dimensional weight space, which is represented as a $m$-dimensional flatten vector. $\mathcal{D}$ denotes the data distribution of $(\mathbf{x}, y)$ and $\mathcal{D}_0/\mathcal{D}_1$ denotes the distribution of the data with sensitive attribute $0/1$. In this work, we consider the fair binary classification ($y \in \{0, 1\}$) with binary sensitive attributes ($s \in \{0, 1\}$).

**Algorithmic Fairness.** For simplicity of the presentation, we focus on group fairness, demographic parity (DP) [3]. The DP metric $\Delta$DP is defined as the absolute difference in the positive prediction rates between the two demographic groups. One relaxed metric $\Delta$DP to measure DP has also been proposed by Edwards and Storkey [21] and widely used by Chuang and Mroueh [23], Agarwal et al. [29], Wei et al. [30], Taskesen et al. [31], Madras et al. [32], which is defined as $\Delta\text{DP}(f) = |\mathbb{E}_{\mathbf{x} \sim \mathcal{D}_0} f(\mathbf{x}) - \mathbb{E}_{\mathbf{x} \sim \mathcal{D}_1} f(\mathbf{x})|$. One practical and effective way to achieve demographic parity is to formulate a penalized optimization with $\Delta$DP as a regularization term in the loss function, which is

$$\mathcal{L} = \mathcal{L}_{ce}(f(\mathbf{x}; \theta), y) + A \cdot \mathcal{L}_f(f(\mathbf{x}; \theta), y) = \mathcal{L}_{ce} + A \cdot \mathcal{L}_f, \tag{1}$$

where $\mathcal{L}_{ce}$ is the objective function (e.g., cross-entropy) of the downstream task, $\mathcal{L}_f$ is instantiated as the demographic parity difference $\Delta\text{DP}(f) = |\mathbb{E}_{\mathbf{x} \sim \mathcal{D}_0} f(\mathbf{x}) - \mathbb{E}_{\mathbf{x} \sim \mathcal{D}_1} f(\mathbf{x})|$, and $A$ [2] is a fixed hyperparameter to balance the model accuracy and fairness. In the following, we set $A$ to 1 for simplicity. We also conducted experiments to explore the effect of varying $A$ in Appendix C.7. In addition to DP, we also consider the fairness metrics of Equality of Opportunity (EO) and Equalized Odds (Eodd) in our experiments, as described in Section 4.6 and Appendix C.6.

**Neural Network Subspaces.** The optimization of the neural network is to find a minimum point (often a local minimum) in a high-dimensional weight space. Wortsman et al. [33] and Benton et al. [34] proposed a method to learn a functionally diverse neural network subspace, which is parameterized by two sets of network weights, $\omega_1$ and $\omega_2$. Such a network will find a minimum "line" The network sampled from the line defined by this pair of weights, $\theta = (1 - \alpha)\omega_1 + \alpha\omega_2$ for $\alpha \in [0, 1]$ Different from this method that the learning objective of both $\omega_1$ and $\omega_2$ are model accuracy for the downstream task, our proposed method lets the $\omega_1$ and $\omega_2$ learn accuracy and fairness, respectively.

## 3. You Only Debias Once

This section introduces our method YODO. Our goal is to find an objective-diverse subspace (the "line") in the weight space comprised of accuracy-optimum (✖ in Figure 1) and fairness-optimum (★ in Figure 1) neural networks, as illustrated in Figure 1. Specifically, we first parameterize two sets of trainable weights $\omega_1 \in \mathbb{R}^n$ and $\omega_2 \in \mathbb{R}^n$ for one neural network. We then optimize weights $\omega_1, \omega_2$, such that $f(\mathbf{x}; \omega_1)$ contributes towards maximizing model accuracy, while $f(\mathbf{x}; \omega_2)$ ensures high model fairness. Thus $f(\mathbf{x}; (1 - \alpha)\omega_1 + \alpha\omega_2))$ achieves $\alpha$-controlled accuracy-fairness trade-offs.

In the training process, we aim to optimize $\omega_1, \omega_2$ with objective functions targeting accuracy and fairness objectives, respectively. The learned $\omega_1, \omega_2$ will be accuracy-optimum (✖ in Figure 1) and fairness-optimum (★ in Figure 1) points in the weight space. To do so, we minimize the loss for the

---

[2] We note the distinction between $A$ and $\alpha$ here. $A$ controls the strength of the fairness regularization of the fairness-optimum model (✖ in Figure 1). $\alpha$ controls the mixing ratio between the accuracy-optimum model (✖ in Figure 1) and the fairness-optimum model (★ in Figure 1). More explanations are presented in Appendix H.

linear combination of two endpoints, which is

$$\mathcal{L} = (1 - \alpha) \underbrace{\mathcal{L}_{ce}}_{\omega_1(\boldsymbol{\times})} + \alpha \underbrace{(\mathcal{L}_{ce} + \mathcal{L}_f)}_{\omega_2(\bigstar)} = \mathcal{L}_{ce} + \alpha \mathcal{L}_f, \tag{2}$$

where $\theta = (1 - \alpha)\omega_1 + \alpha\omega_2$ for $\alpha \in [0, 1]$. $\mathcal{L}_{ce}$ is instantiated as the cross-entropy loss for downstream tasks (i.e., binary classification task), and $\mathcal{L}_f$ is instantiated as demographic parity difference $\Delta\mathrm{DP}[f(x;\theta)]$. Since we seek to optimize the different levels of fairness constraints, for each $\alpha \in [0, 1]$, we propose to minimize $\mathbb{E}_{(\mathbf{x},y)\sim\mathrm{D}}[\mathcal{L}_{ce}(f(\mathbf{x};\theta),y) + \alpha\mathcal{L}_f(f(\mathbf{x};\theta),y)]$, thus objective is to minimize

$$\mathbb{E}_{\alpha\sim\mathrm{U}}\big[\mathbb{E}_{(\mathbf{x},y)\sim\mathrm{D}}[\mathcal{L}_{ce}(f(\mathbf{x};\theta),y) + \alpha\mathcal{L}_f(f(\mathbf{x};\theta),y)]\big], \quad s.t. \quad \theta = (1 - \alpha)\omega_1 + \alpha\omega_2, \tag{3}$$

where U denotes the uniform distribution between 0 and 1 and the trade-off hyperparameter $\alpha$ follows uniform distribution, i.e., $\alpha \sim \mathrm{U}$.

**Prediction Procedure.** After training a model $f(\mathbf{x};\omega_1,\omega_2,\alpha)$ with two sets of parameters $\omega_1$ and $\omega_2$, the prediction procedure for a test sample $\mathbf{x}$ is *i*) Choose the desired trade-off parameter $\alpha$, which controls the balance between accuracy and fairness, *ii*) Compute the weighted combination of the two sets of trained weights, $(1 - \alpha)\omega_1 + \alpha\omega_2$, to obtain the model parameters for the desired trade-off, *iii*) Compute the prediction function to the test sample $\mathbf{x}$ as $f(\mathbf{x};(1 - \alpha)\omega_1 + \alpha\omega_2)$, to

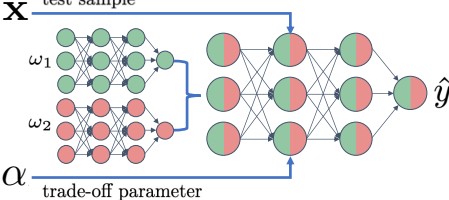

Figure 2: Prediction procedure of YODO

obtain the predicted output. This prediction procedure offers flexible accuracy-fairness trade-offs at inference time, enabling users to choose the desired level of trade-offs for specific applications.

**Why Does** YODO **Achieve Flexible Accuracy-fairness Trade-offs?** Since our method aims to find the "line" between the accuracy-optimum and fairness-optimum points in the weight space, we need to ensure that any point on this line corresponds to a specific level of fairness. Therefore, we randomly sample a $\alpha$ during each batch. For each $\alpha$ in U, the objective $\mathbb{E}_{(\mathbf{x},y)\sim\mathrm{D}}[\mathcal{L}_{ce}(f(\mathbf{x};\theta),y) + \alpha\mathcal{L}_f(f(\mathbf{x};\theta),y)]$ will be optimized to the minima, leading to different accuracy-fairness trade-offs with different $\alpha$. In other words, each $\alpha$ corresponds to one model with a specific level of fairness. Under a wide range of different $\alpha$ values, we train numerous models throughout the training process. Such a mechanism guarantees that YODO could achieve flexible trade-offs with one-time training at inference time. We also provide the analysis for YODO from the gradient perspective in Appendix A.

**Model Optimization.** To promote the diversity of the neural network subspaces, we follow the approach proposed in [33] and aim to learn two distinct endpoints, i.e., the accuracy-optimum and the fairness-optimum endpoint in the weight space. To achieve this, we add a cosine similarity regularization term $\mathcal{L}_{reg} = \frac{\langle\omega_1,\omega_2\rangle^2}{|\omega_1|_2^2|\omega_2|_2^2}$, which encourages the two endpoints to be as dissimilar as possible. Minimizing $\mathcal{L}_{reg}$ promotes diversity between the two endpoints. Thus the final objective function will be $\mathcal{L} = \mathcal{L}_{ce} + \alpha\mathcal{L}_f + \beta\mathcal{L}_{reg}$. The algorithm of YODO is presented in Algorithm 1.

**Model Complexity.** we analyze the time and space complexity of YODO. Compared to the memory utilization of the models with fixed accuracy-fairness trade-off, our method utilizes twofold memory usage. However, it can achieve in-situ flexible trade-offs at inference time. At the training time, the computational cost is from gradient computation of $\omega_1/\omega_2$ (Equation (5)). We also conducted experiments comparing the fixed model and YODO to evaluate the additional running time and presented the results in Table 1. The results show that, on average, YODO only results in a 36% increase in training time. When compared to an arbitrary accuracy-fairness

Table 1: Comparing the running time for the fixed-training model and the YODO, conducted on an NVIDIA RTX A5000. The results are the mean of 10 trials. The unit is second.

| Datasets | Fixed | YODO | Extra Time |
|---|---|---|---|
| UCI Adult | 0.42 | 0.58 | 38% |
| KDD Census | 3.78 | 4.99 | 32% |
| ACS-I | 3.93 | 6.09 | 55% |
| ACS-E | 3.53 | 4.25 | 20% |
| Average | 2.91 | 3.97 | 36% |

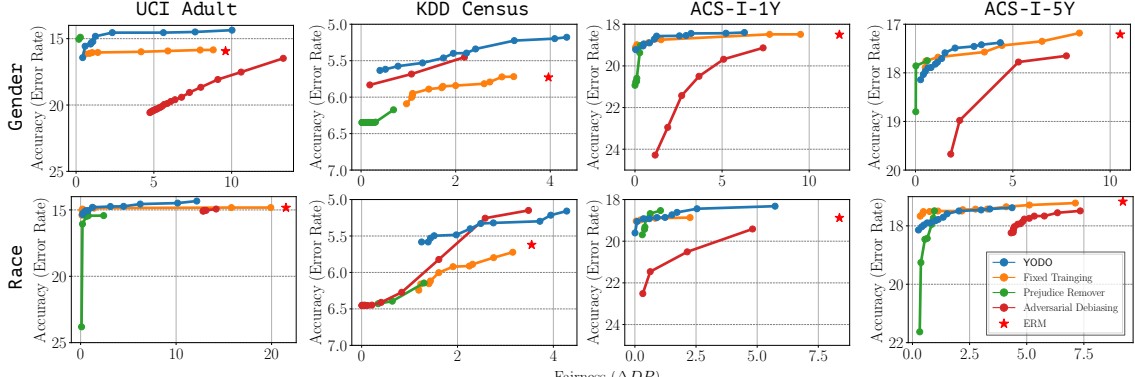

Figure 3: The Pareto frontier of accuracy and fairness. The first row is the fairness performance with respect to gender sensitive attribute, while the second row is race sensitive attribute. The model performance metric is Error Rate (lower is better), and the fairness metric is ΔDP (lower is better).

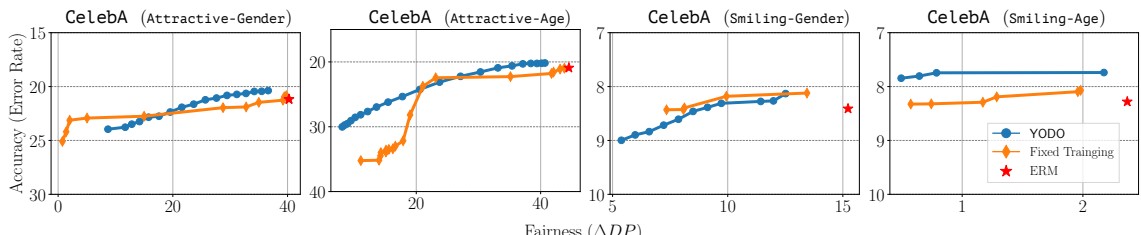

Figure 4: The Pareto frontier of the model performance and fairness on the CelebA dataset. The sensitive attribute we considered is gender and age. The x-axis represents the ΔDP, while the y-axis represents the error rate of the downstream task. Our proposed one-time training model achieves a comparable accuracy-fairness trade-off with that of fixed-trained models.

trade-off, this extra time is negligible. For example, if we need 100 levels of trade-off on the ACS-E dataset, YODO takes 3.53 seconds, while training 100 fixed models takes 425 seconds. In this sense, the 3.53 seconds needed by YODO is considered negligible.

## 4. Experiments

We empirically evaluate the performance of our proposed method. *For dataset*, we use both tabular and image data, including UCI Adult [35], KDD Census [35], ACS-I (ncome) [36], ACS-E (mployment) [36] as tabular data, and CelebA as image dataset. *For baselines*, we mainly include **ERM** and **Fixed Training**. Fixed Training trains multiple models for different fixed accuracy-fairness trade-offs. For each point in Figures 3 and 12, we trained one fixed model using the objective function shown in Equation (1) with different values of $A$, which is set to $(0, 1]$ with an interval of 0.05. In addition to that, we also consider more baseline methods, including Prejudice Remover [37], Adversarial Debiasing [22] for demographic parity. [3] We provide more details of baselines in Appendix D.2. For our experimental setting, we have to train 20 individual models from scratch for each trade-off hyperparameter. Another baseline is Empirical Risk Minimization (ERM), which is to minimize the empirical risk of downstream tasks (marked as ★ in the figures). The additional experimental settings are presented in Appendix C. The major **Obs**ervations are highlighted in **boldface**.

### 4.1. Will YODO **Achieve Flexible Trade-offs only with Training Once?**

We validate the effectiveness of our proposed YODO on real-world datasets, including tabular data and image data. We present the results in Figures 3, 4 and 12 and use Pareto frontier [38] to evaluate our proposed method and the baseline. Pareto frontier is widely used to evaluate the accuracy-fairness

---

[3]We note that all these baselines use fixed training (i.e., each model represents a single level of fairness), while our proposed YODO trains once to achieve a flexible level of fairness.

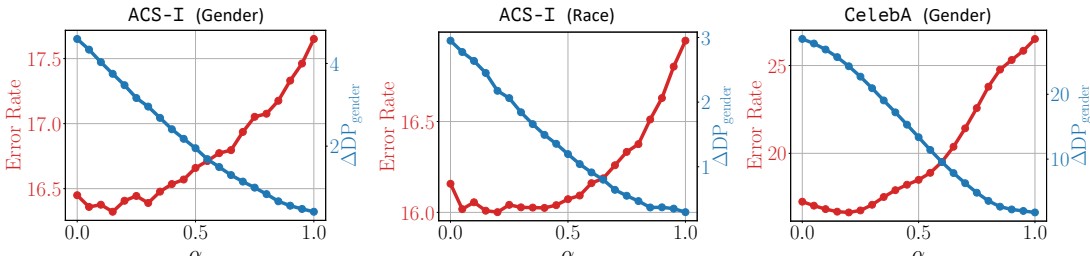

Figure 5: The accuracy-fairness trade-offs at inference time with respect to $\alpha$ for three different datasets: `ACS-I` dataset with gender as the sensitive attribute (**Left**), `ACS-I` dataset with race as the sensitive attribute (**Middle**), `CelebA` dataset with gender as the sensitive attribute (**Right**). We observed that the fine-grained accuracy-fairness trade-offs could be achieved by selecting different values of $\alpha$, providing more nuanced accuracy-fairness trade-offs. Note that the results are obtained at inference time with a single trained model.

trade-offs by Kim et al. [39], Liu and Vicente [40], Wei and Niethammer [41] and characterizes the achievable accuracy of a model for given fairness conditions. Pareto frontier characterizes a model's achievable accuracy for given fairness conditions, as a measurement to understand the trade-offs between model accuracy and fairness. The details of the experiments can be found at Appendix D.4. The results on the tabular dataset are presented in Figures 3 and 12. The results on the image dataset are presented in Figure 4. From these figures, we make the following major observations:

**Obs.1: Even though** `YODO` **only needs to be trained once, it performs similarly to the baseline (Fixed Training), or even better in some cases.** We compared the Pareto frontier of `YODO` with that of the **Fixed Training** baseline. We found that the Pareto frontier of `YODO` coincides with that of **Fixed Training** in most figures, indicating it can achieve comparable or even better performance than the baseline. The result makes `YODO` readily usable for real-world applications requiring flexible fairness. It is worth highlighting that our proposed method only needs one-time training to obtain the Pareto frontier at inference time, making it computationally efficient.

**Obs.2: On some datasets (e.g., `UCI Adult`, `KDD Census`, `CelebA`),** `YODO` **even outperforms the ERM baseline.** Upon comparing our results on the `UCI Adult` and `KDD Census` datasets, we observed that the Pareto frontier of our method covers the point of ERM. This finding suggests that our approach outperforms ERM in both model accuracy and fairness performance. It also demonstrates that our proposed objective-diverse neural network subspace can achieve flexible accuracy-fairness trade-offs while potentially improving model accuracy. This improved performance comes from our method's ability to learn a richer and larger space (a line in the weight space) rather than the fixed trained model (a point in the weight space).

**Obs.3: `YODO` likely achieves a smoother Pareto frontier than baseline (Fixed Training).** On most datasets, especially on `CelebA` image data, our proposed method demonstrates a smoother Pareto frontier compared to the baselines. For example, on `CelebA` Attractive-Age and Attractive-Gender in Figure 4, the Pareto frontier is notably smoother than baselines. A smooth Pareto frontier implies that our proposal is able to achieve more fine-grained accuracy-fairness trade-offs and the smoothness of the fairness-accuracy trade-off curve implies the model's stability and robustness.

## 4.2. Can Accuracy-fairness Trade-Offs Be Flexible by Controlling Parameter $\alpha$?

To explore the effect of hyperparameter $\alpha$ on the accuracy-fairness trade-offs, we conducted experiments on the `ACS-I` and `CelebA` datasets with different values of $\alpha$ at inference time. $\alpha$ is a hyperparameter that controls the trade-off between model accuracy and fairness in machine learning models. A higher value of $\alpha$ emphasizes fairness, while a lower value prioritizes accuracy. The results are presented in Figure 5 and we observed:

**Obs.4: The accuracy-fairness trade-offs can be controlled by $\alpha$ at inference time.** Our experiments on both tabular and `CelebA` datasets showed that as we increase the value of the hyperparameter $\alpha$,

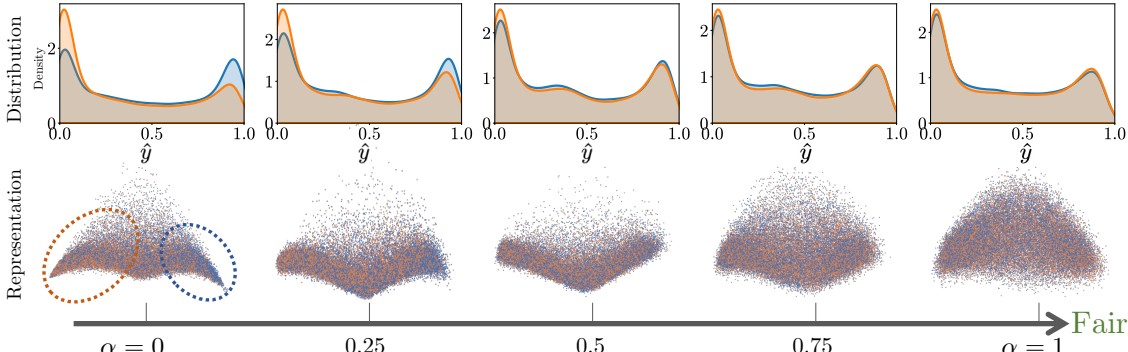

Figure 6: The changing distribution of the prediction values $\hat{y}$ and the representations as $\alpha$ increases. Blue indicates male and orange indicates female. **Top**: The distribution is estimated with kernel density estimation [42]. The distributions are more polarized between males and females with $\alpha = 0$, but become more similar with $\alpha = 1$. As $\alpha$ increases, the distributions for male and female groups become more similar, indicating achieving demographic parity. **Bottom**: The visualizations of the representation with the values of $\alpha$ are from 0 to 1. The figure at the far left ($\alpha = 0$) shows that the representations for males (⋯) and females (⋯) are distinctly separate, indicating that the representations contain more sensitive information. The figure at the far right ($\alpha = 1$) shows that the representations of different groups are mixed together, indicating less sensitive information.

the error rate (lower is better) gradually increases while $\Delta$DP gradually decreases. The solution at $\alpha = 0$ prioritizes accuracy-optimum (lowest error rate), while the solution at $\alpha = 1$ prioritizes fairness-optimum (lowest $\Delta$DP). These results demonstrate that our proposed method can achieve flexible accuracy-fairness trade-offs.

### 4.3. How Does YODO Behave with Changing $\alpha$ ?

We examine the distribution of predictive values and the hidden representation to investigate the varying trade-offs. We specifically plot the distribution of predictive values for different groups (male and female) to verify the flexible accuracy-fairness trade-offs provided by our approach. With the different combinations with different $\alpha$s, we visualize the hidden representation [43, 44] in Figure 6. The experiments are conducted on ACS-I dataset, and the results are presented in Figure 6.

**Obs.5: The distribution of predictive values for different groups becomes increasingly similar as the value of $\alpha$ increases**, indicating that our model becomes more fair as the values of $\alpha$ increase. Additionally, we found that the distributions of the predictive values of different groups follow the same distribution, showing the predictive values are independent of sensitive attributes. The varying distribution of predictive values provides valuable insights into why our model can achieve flexible accuracy-fairness trade-offs from a distributional perspective.

**Obs.6: The disparity of the representation of different groups becomes smaller and smaller with the increasing $\alpha$.** The figure on the far left in Figure 6 shows that the representation of male and female groups are distinctly separate, indicating that the representations contain more sensitive information. The figure on the far right shows the representations of different groups mixed together, indicating that the representations contain little sensitive information. The sensitive information in the representation indicates that the two endpoints correspond to accuracy and fairness, respectively.

### 4.4. How Does YODO Perform with Different the Balance Parameter $A$?

In this experiment, we evaluated the performance of YODO with varying balance parameter values ($A$). We tested the model performance with varying $A$, ranging from 1 to 5, with increments of 0.5. We present the results in Figure 7. The results show that YODO exhibited its best performance when $A = 1$, as this specific balance parameter value resulted in higher accuracy and a larger demographic parity span compared to other values of $A$. As the value of $A$ increases, although the fairness performance improves, the accuracy of the downstream task deteriorates. When $A = 5$, the error rate of the downstream task is even worse than the highest error rate of the model with $A = 1$. The poorer performance for the downstream task demonstrates that the model with $A = 5$ is inferior

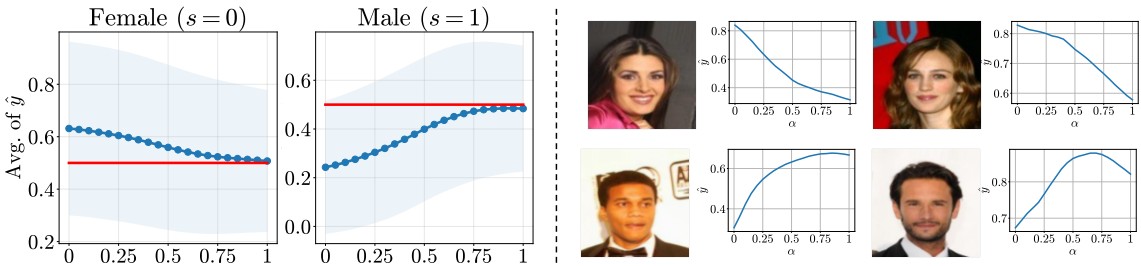

Figure 8: Case study on `CelebA` dataset. The sensitive attribute is gender. The downstream task is to predict whether a person is attractive or not. The results show that the YODO can provide the instance-level prediction change for practitioners to examine the fairness performance.

to that with $A = 1$. We also observed that setting $A = 1$ effectively addresses the trade-off between accuracy and fairness, achieving an optimal balance. More details are presented in Figures 17 and 18.

### 4.5. How Does the Prediction Value Vary at the Instance Level? A Case Study

We experiment on the `CelebA` dataset to investigate how the predicted values change and the prediction with the various $\alpha$ in the instance level. The sensitive attribute we consider is Gender. The downstream task is to predict whether a person is attractive or not. The female group has more positive samples than the Male group, leading the biased prediction. We also present some cases to investigate the effect of the change of the values of $\alpha$ in Figure 8. Based on our analysis of Figures 8 and 11, we make the following observations:

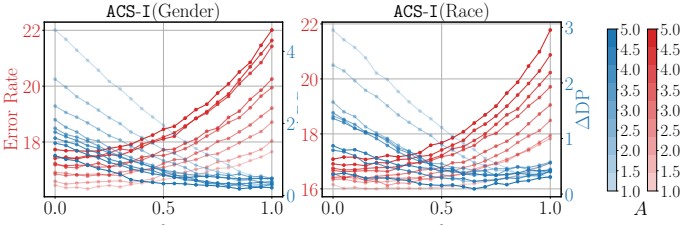

Figure 7: The effect of the accuracy-fairness balance parameter $A$. The $\alpha$ in the x-axis controls the accuracy-fairness trade-off at inference time. The strength of the color reflects the value of $A$. The optimal balance occurs at around $A = 1$.

**Obs.7:** YODO **can provide an instance-level explanation for group fairness with only one model, while fixed training models need multiple models.** In `CelebA` dataset, the Female group has more positive (attractive) samples than the other one, leading to a higher predictive value for the male group. Figure 8 shows that YODO tends to lower the predictive values of the Female group while increasing the predictive values of the Male group. Such a tendency would result in fairer predictive results. Our proposed method YODO offers an instance-level explanation for individuals with only one model. For example, Figure 8 demonstrates how YODO can lower the predictive values for the Female group and increase the predictive values for the Male group, resulting in a fairer outcome for individuals belonging to each group. Our method can provide individualized explanations and a trustworthy model for end-users with only one model.

### 4.6. How Does YODO Perform on Other Group Fairness?

In this section, we experiment on the fairness metric Equality of Opportunity (EO) and Equalized Odds (Eodd) [45]. EO requires that the True Positive Rate (TPR) for a predictive model be equal across different groups. And a classifier adheres to Eodd if it maintains equal true positive rates and false positive rates across all demographic groups. The results are presented in Figures 9, 10 and 13 and more details are presented in Appendix C.6. We have the following observations:

**Obs.8:** YODO **performs similarly to baseline (Fixed Training) in terms of Equality of Opportunity, or even better in some cases.** This observation indicates it can achieve comparable or even better performance than the baseline. And this experiment also makes our method readily usable for real-world applications, which require flexible fairness. The result shows the effectiveness of our proposal on other fairness metrics, demonstrating its overall effectiveness in promoting fairness.

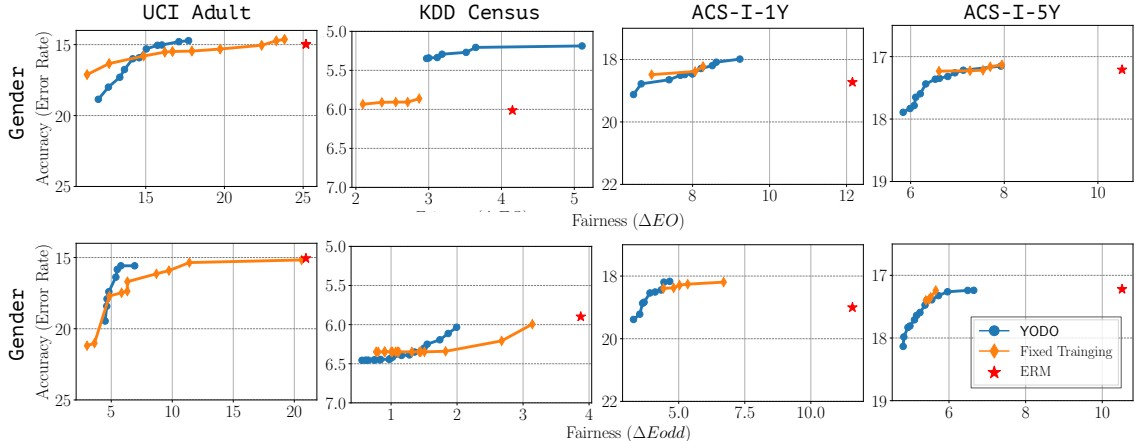

Figure 9: The Pareto frontier of accuracy and fairness. The model performance metric is Error Rate (lower is better), and the fairness metric are ΔEO and ΔEodd (lower is better).

# 5. Related Works

**Fairness in Machine Learning.** Recently, algorithmic fairness [46–59] is required legally or morally in machine learning systems and various fairness definitions in machine learning systems have been proposed to meet the requirement of fairness expectation. The fairness can typically be classified into *individual fairness* or *group fairness*, which can be achieved via pre/in/post-processing. In this paper, we focus on the in-processing group fairness, which measures the statistical parity between subgroups defined by the sensitive attributes, such as gender or race [23, 32, 44, 45, 60–62]. Nevertheless, these constraints are trained and satisfied during training, the model may expect different accuracy-fairness trade-offs at inference time. In contrast, Our proposed method, YODO, aims to address this issue by enabling flexible accuracy-fairness trade-offs at inference time.

**Accuracy-Fairness Trade-offs.** Many existing works investigate the trade-offs between the model accuracy and fairness [17, 18, 39–41, 63–71]. Maity et al. [69] discusses the existence of the accuracy-fairness trade-offs. [18, 39, 39, 68, 68, 72–75, 75, 76] shows that fairness and model accuracy conflict with each another, and achieved fairness often comes with a necessary cost in loss of model accuracy. Kim et al. [39], Cooper et al. [63] re-examines the trade-offs and concludes that unexamined assumptions may result in emergent unfairness.

**Neural Network Subspaces Learning.** The idea of learning a neural network subspace is concurrently proposed by Wortsman et al. [33] and Benton et al. [34]. Multiple sets of network weights are treated as the corners of a simplex, and an optimization procedure updates these corners to find a region in weight space in which points inside the simplex correspond to accurate networks. Garipov et al. [27] learning a connection between two independently trained neural networks, considering curves and piecewise linear functions with fixed endpoints. Wortsman et al. [33] and Benton et al. [34] concurrently proposed to learn a functionally diverse subspace. Our proposed method YODO differs from these methods by specifying each subspace to the different learning objectives and allowing flexible fairness levels at inference time.

# 6. Conclusion

In this paper, we proposed YODO, a novel method that achieves accuracy-fairness trade-offs at inference time to meet the diverse requirements of fairness in real-world applications. Our approach is the first to achieve flexible trade-offs between model accuracy and fairness through the use of an objective-diverse neural network subspace. Our extensive experiments demonstrate the effectiveness and practical value of the proposed approach, and we offer a detailed analysis of the underlying mechanisms by examining the distributions of predictive values and hidden representations. By enabling in-situ flexibility, our approach can provide more nuanced control over the trade-offs between accuracy and fairness, thereby advancing the state-of-the-art in the field of fairness.

# Acknowledgement

We thank the anonymous reviewers for their constructive suggestions and fruitful discussion. Portions of this research were conducted with the AISC at Case Western Reserve University supported by NSF-2117439. This work is, in part, supported by CNS-2431516, NIH under other transactions 1OT2OD038045-01, NSF IIS-2224843, IIS-2310260, NIH Aim-Ahead program. The views and conclusions contained in this paper are those of the authors and should not be interpreted as representing any funding agencies.

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

# Appendix

## Table of Contents

# A. Optimization Resulting in Objective-Diverse Subspace

We discuss the model optimization for YODO from the gradient perspective. In each training batch, we randomly sample $\alpha \sim \mathrm{U}_{[0,1]}$, and then we use $\theta = (1 - \alpha)\omega_1 + \alpha\omega_2$ as the model parameters. We calculate the gradients for $\omega_1$ and $\omega_2$ with respect to objective function (Equation (3)) as follows:

$$\frac{\partial \mathcal{L}}{\partial \omega_i} = \frac{\partial \mathcal{L}}{\partial \omega_i} = \frac{\partial \mathcal{L}}{\partial \theta}\frac{\partial \theta}{\partial \omega_i}. \tag{4}$$

Consider that $\theta = (1 - \alpha)\omega_1 + \alpha\omega_2$, the gradients for the endpoints $\omega_1$ and $\omega_2$ are

$$\frac{\partial(\mathcal{L}_{ce} + \alpha\mathcal{L}_f)}{\partial \omega_1} = (1 - \alpha)\frac{\partial(\mathcal{L}_{ce} + \alpha\mathcal{L}_f)}{\partial \theta}, \qquad \frac{\partial(\mathcal{L}_{ce} + \alpha\mathcal{L}_f)}{\partial \omega_2} = \alpha\frac{\partial(\mathcal{L}_{ce} + \alpha\mathcal{L}_f)}{\partial \theta}, \tag{5}$$

From Equations (4) and (5), we can see that gradients for $\omega_1$ and $\omega_2$ are related to the $\mathcal{L}_{ce}$ and $\mathcal{L}_f$ with different values of the scale coefficient (i.e., $1 - \alpha$ and $\alpha$). The optimization of the two endpoints $\omega_1$ and $\omega_2$ only depends on the gradient $\frac{\partial(\mathcal{L}_{ce} + \alpha\mathcal{L}_f)}{\partial \theta}$, which are the most time-consuming operations in the back-propagation during the model training. This indicates that our method does not require any extra cost in the back-propagation phase since we only compute $\frac{\partial(\mathcal{L}_{ce} + \alpha\mathcal{L}_f)}{\partial \theta}$ once.

**Why Does** YODO **Achieve Flexible Accuracy-fairness Trade-offs in Terms of Gradient?** Equation (5) indicates the optimization of the weights $\omega_1$ and $\omega_2$, as well as the objective function, are both controlled by the hyperparameter $\alpha$. In the following analysis, we will examine the optimization of these weights with different values of $\alpha$.

- When $\alpha = 0$, the gradients of $\omega_1$ in Equation (5) is calculated by the loss $\frac{\partial(\mathcal{L}_{ce} + \alpha\mathcal{L}_f)}{\partial \omega_1} = (1 - 0)\frac{\partial(\mathcal{L}_{ce} + 0*\mathcal{L}_f)}{\partial \theta} = \frac{\partial(\mathcal{L}_{ce})}{\partial \theta}$ and the gradients of $\omega_2$ is $0$, indicating weight $\omega_1$ will be only optimized by the accuracy objective.

- When $\alpha = 1$, the gradients in Equation (5) is calculated by the loss $\frac{\partial(\mathcal{L}_{ce} + \alpha\mathcal{L}_f)}{\partial \omega_2} = 1 * \frac{\partial(\mathcal{L}_{ce} + 1*\mathcal{L}_f)}{\partial \theta} = \frac{\partial(\mathcal{L}_{ce} + \mathcal{L}_f)}{\partial \theta}$ and the gradients of $\omega_1$ is $0$, indicating weight $\omega_2$ will be only optimized by the fairness objective.

- When $0 < \alpha < 1$, the weights $\omega_1$ and $\omega_2$ will be only optimized by the linear combination of accuracy objective and fairness objective.

From the analysis of gradient, we can conclude that *the optimization tends to encourage the two endpoints $\omega_1$ and $\omega_2$ to accuracy-optimum and fairness-optimum solutions in the weight space*. As illustrated in Figure 1(b), we plot the landscape of the error rate (accuracy) and the $\Delta$DP(fairness). The landscapes are depicted with a trained YODO. ✖ and ★ indicate the two endpoints, which are encouraged to learn accuracy and fairness objectives, respectively. The landscape shows that our method learns an objective-diverse neural network subspace and optimizes the endpoints to accuracy-optimum and fairness-optimum solutions.

# B. The Pseudo-code for YODO

We provide the Pseudo-code for the training and inference of YODO as the following algorithms. The training and prediction procedure for YODO are presented in Algorithms 1 and 2, respectively.

---

**Algorithm 1** Pseudo-code for YODO Training

---

**Require:** Training set $\mathcal{S}$, balance hyperparameters $\beta$
1: Initialize each $\omega_i$ independently.
2: **for** batch in $\mathcal{S}$ **do**
3:      Randomly sample an $\alpha$
4:      Calculate interpolated weight $\omega =\leftarrow (1-\alpha)\omega_1 + \alpha\omega_2$
5:      Calculate loss $\mathcal{L} = \mathcal{L}ce + \alpha\mathcal{L}f + \beta\mathcal{L}_{reg}$
6:      Back-propagate $\mathcal{L}$ to update $\omega_1$ and $\omega_2$ using Equation (5).
7: **end for**

---

**Algorithm 2** Pseudo-code for YODO Prediction

---

**Require:** Training set $\mathcal{S}$, balance hyperparameters $\beta$
1: Initialize each $\omega_i$ independently.
2: **for** batch in $\mathcal{S}$ **do**
3:      Randomly sample an $\alpha$
4:      Calculate interpolated weight $\omega \leftarrow (1-\alpha)\omega_1 + \alpha\omega_2$
5:      Calculate the prediction with the interpolated weight $\hat{y} = f(x; \omega)$
6: **end for**

---

# C. Additional Experiments

In this appendix, we provide addition experimental results. In additional, we carried out supplementary experiments to delve deeper into the analysis of our proposed model. These experiments encompass the following aspects: comparing the training of two separate models, examining Equalized Odds, investigating the impact of varying the accuracy-fairness balance parameter $A$, and evaluating the performance of the YODO model with larger architectures.

## C.1. Experiments on `CelebA` Dataset with $\Delta$EO

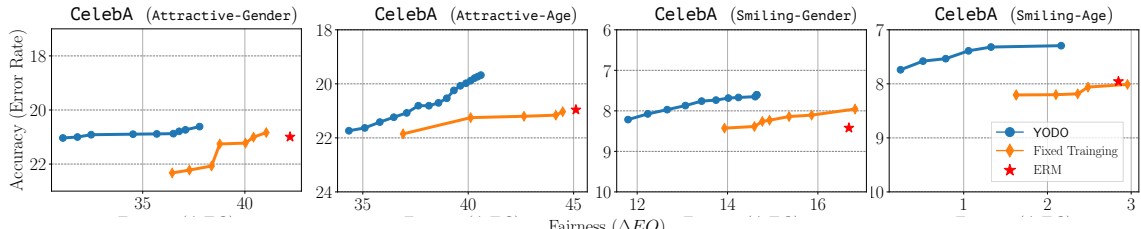

Figure 10: The Pareto frontier of the model performance and fairness on `CelebA` dataset. The downstream task is to predict whether a person is Attractive (Smiling) or not. The sensitive attribute is gender and age.

## C.2. The Mean of the Prediction Values Change with the Change of $\alpha$

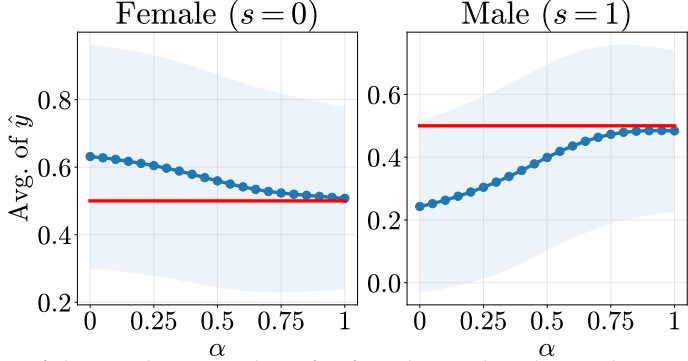

Figure 11: The mean of the prediction values for females and males with respect to varying $\alpha$. The red line indicates the predictive value $\hat{y} = 0.5$. The blue line is the mean of predictive values of each group (i.e., male and female). The results illustrate that the mean prediction values for males and females become more similar as $\alpha$ increases., indicating a fairer prediction.

We observed that the mean of the predictive values of different groups (e.g., female, male) approaches 0.5 with the increasing $\alpha$. Analysis of Figure 8 shows that, overall, YODO tends to decrease the predictive values for the Female group and increase the predictive values for the Male group, resulting in lower $\Delta$DP values and indicating fairer predictions overall. Our observations demonstrate that our proposed method can mitigate the unfairness caused by the dataset's gender imbalance and ensure that individuals from different groups are treated equitably.

## C.3. Experiments on `ACS-E` Dataset

In this appendix, we present the results of complementary experiments conducted on the `ACS-E` dataset with gender as the sensitive attribute. The results for two-group fairness metrics, $\Delta$DP and $\Delta$EO, are presented in Figures 12 and 13, respectively. These results are complementary to the ones presented in Figures 3 and 9.

The experimental results on the `ACS-E` dataset with gender as the sensitive attribute show that our proposed method achieves comparable accuracy-fairness trade-offs to the baselines for both group fairness metrics $\Delta DP$ and $\Delta EO$. This finding is consistent with the results of the experiments on other datasets with different sensitive attributes presented in Figures 3 and 9.

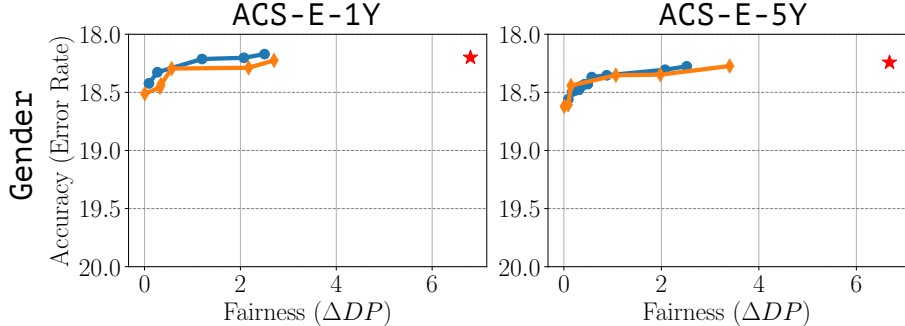

Figure 12: The Pareto frontier of the model performance and fairness on `ACS-E` dataset spanning 1 year or 5 years on $\Delta DP$ metric. The sensitive attribute is gender.

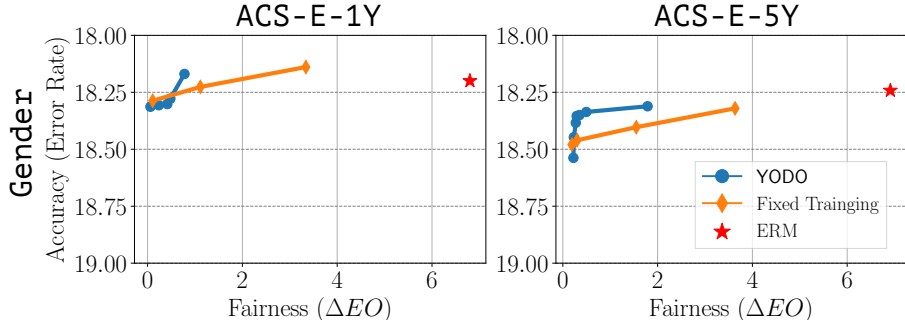

Figure 13: The Pareto frontier of the model performance and fairness on `ACS-E` data spanning 1 year or 5 years on $\Delta EO$ metric. The sensitive attribute is gender.

## C.4. The Distribution of Predictive Values on Image Dataset

In this appendix, we provide the distribution of the predictive values on the image dataset, `CelebA`. The distribution of the predictive values of different groups are more and more similar with the increase of the values of $\alpha$, indicating that our model is more and more fair with the increase of the values of $\alpha$. This result shows that our proposed method encourages the distribution to follow the same distribution and further guarantees fair prediction.

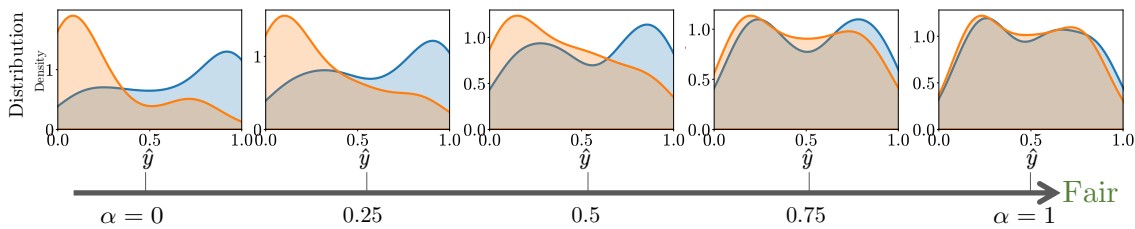

Figure 14: The distribution of the prediction values with different $\alpha$ on `CelebA` dataset. The distribution of the predictive values of different groups (i.e., Male, Female) becomes more and more similar with the increasing $\alpha$. The distributions are more polarised between Male and Female with $\lambda = 0$ while the distribution is nearly the same with $\lambda = 1$.

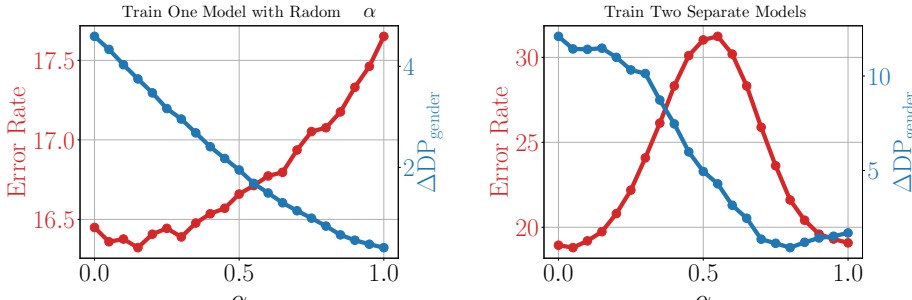

Figure 15: Comparison of Two Training Strategies. **Left:** Training a single model with two sets of parameters $\omega_1$ and $\omega_2$. **Right:** Training two separate models with $\omega_1$ and $\omega_2$. The error rate in the right subfigure shows that interpolating the weights $\omega_1$ and $\omega_2$ of the two well-trained models at inference time achieves no worse accuracy, since the error rate is relatively high when $\alpha = 0.5$.

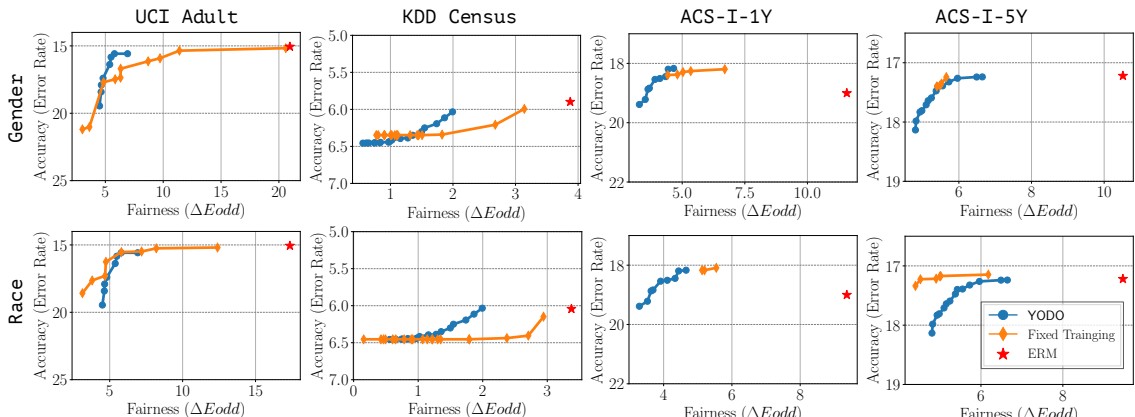

Figure 16: The Pareto frontier of the model accuracy and fairness. The first row is the fairness performance with respect to gender sensitive attribute, while the second row is race sensitive attribute. The model performance metric is Error Rate (lower is better), and the fairness metric is $\Delta$Eodd (lower is better).

## C.5. Comparison to Training Two Separate Models

Training two models separately and interpolating them can not achieve our goal- flexible accuracy-fairness trade-offs at inference time. As verified and studied by previous works Wortsman et al. [33], Benton et al. [34], Frankle et al. [77], Fort et al. [78], interpolating the weights $\omega_1$ and $\omega_2$ of two well-trained models at inference time has been shown to achieve no better accuracy than an untrained model. We also conduct experiments to verify this point in our case with two settings: 1) training a single model with two sets of parameters $\omega_1$ and $\omega_2$. 2) Training two separate models with $\omega_1$ and $\omega_2$. We present the results in Figure 15. The error rate in the right subfigure shows that interpolating the weights $\omega_1$ and $\omega_2$ of the two well-trained models at inference time achieves no worse accuracy since the error rate is relatively high when $\alpha = 0.5$.

## C.6. Experiments on Equalized Odds

In this section, we experiment on the fairness metric Equalized Odds (Eodd) [45]. Eodd assesses whether a classifier offers equal opportunities to individuals from diverse groups. A classifier adheres to this criterion if it maintains equal true positive rates and false positive rates across all demographic groups. We present a relaxed version of Eodd as follows:

$$\Delta\text{Eodd}(f) = |\mathbb{E}_{\mathbf{x}\sim\mathcal{D}_0, y=1}f(\mathbf{x}) - \mathbb{E}_{\mathbf{x}\sim\mathcal{D}_1, y=1}f(\mathbf{x})| + |\mathbb{E}_{\mathbf{x}\sim\mathcal{D}_0, y=0}f(\mathbf{x}) - \mathbb{E}_{\mathbf{x}\sim\mathcal{D}_1, y=0}f(\mathbf{x})|. \quad (6)$$

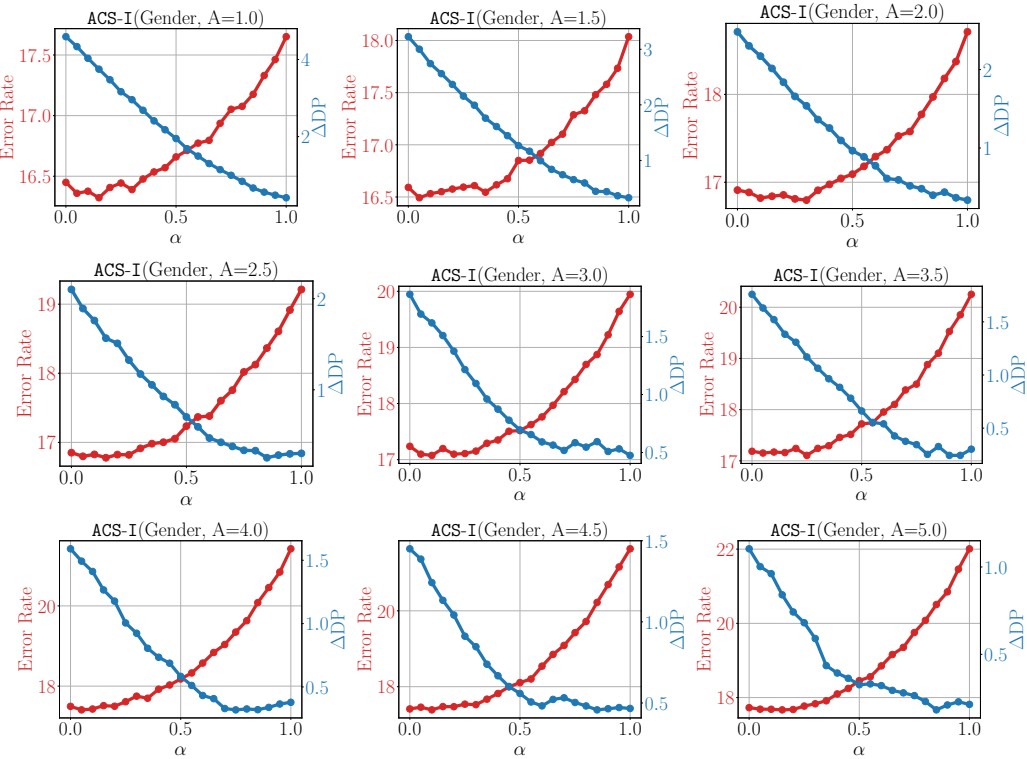

Figure 17: The effect of the accuracy-fairness balance parameter $A$ on ACS-I dataset with Gender as the sensitive attribute. The results show that our method can achieve flexible accuracy-fairness trade-offs at inference time with all the values of $A$.

We observed that YODO **performs similarly to baseline (Fixed Training) in terms of Equalized Odds, or even better in some cases.** This observation indicates our proposed method can achieve comparable or even better performance than the baseline. The result shows the effectiveness of our proposal on other fairness metrics, demonstrating its overall effectiveness in promoting fairness.

## C.7. Impact of Different Values of Accuracy-fairness Balance Parameter $A$

In addition to the combined results for different $A$ values presented in Figure 7, we also display the results for each individual $A$ value in separate figures, as shown in Figure 17 and Figure 18. The results demonstrate that models with various $A$ values can achieve a range of accuracy-fairness trade-offs. However, as $A$ increases, the overall accuracy of the downstream task declines while the span of the fairness metric $\Delta$DP expands.

## C.8. The Performance of YODO with Larger Models

We conducted experiments on larger models using the CelebA dataset, specifically examining ResNet18, ResNet34, ResNet50, ResNet101, WideResNet50, and WideResNet101 [79, 80]. We investigated the accuracy-fairness trade-off behavior of these models while varying the $\alpha$ parameter. We note that all the results are referenced with one model at the inference time. Our observations revealed that 1) All models were capable of achieving a flexible accuracy-fairness trade-off during inference. 2) Larger models demonstrated greater stability, with smaller models like ResNet18 and ResNet34 exhibiting more fluctuations, while larger models such as ResNet50, ResNet101, WideResNet50, and WideResNet101 showed smoother performance.

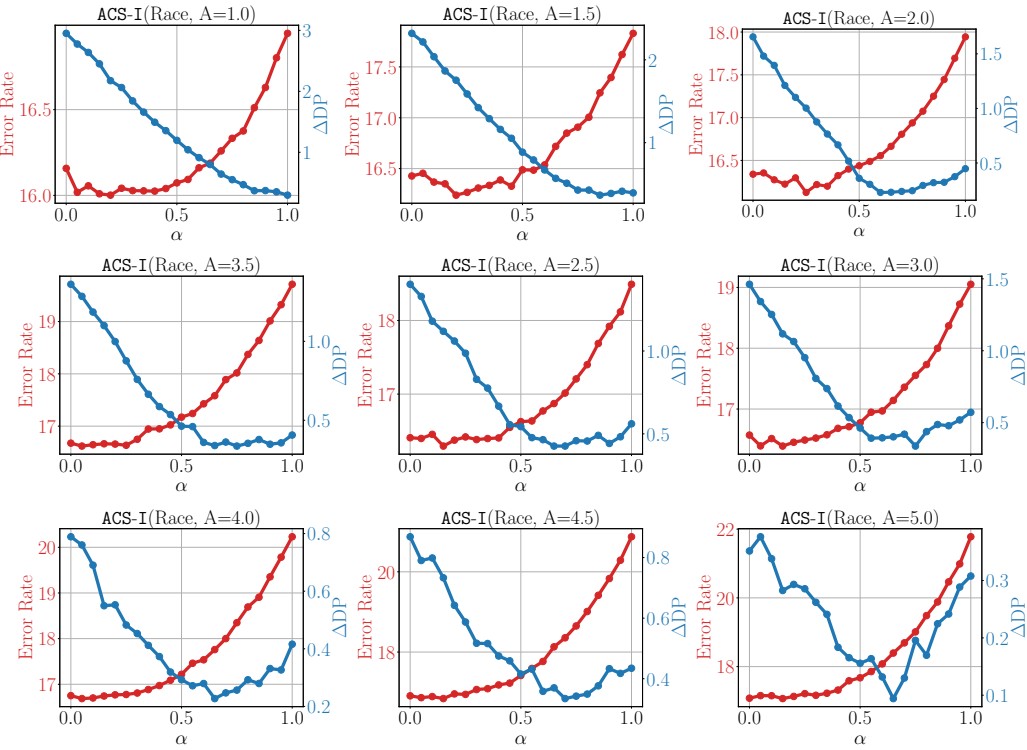

Figure 18: The effect of the accuracy-fairness balance parameter $A$ on ACS-I dataset with Race as the sensitive attribute. The results show that our method can achieve flexible accuracy-fairness trade-offs at inference time with all the values of $A$.

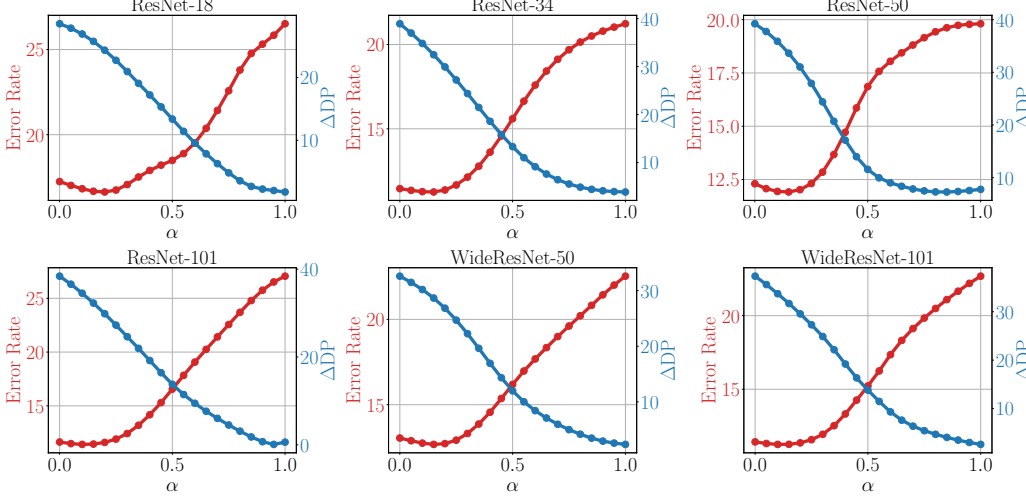

Figure 19: Performance of YODO with larger models.

# D. Experiment Details

This section provides a detailed description of our experiment setting, including the description of the datasets, neural network architectures, and experiment settings used in our experiments.

## D.1. Dataset

In the appendix, we provide more details about the datasets used in the experiments. These include tabular datasets such as `UCI Adult`, `KDD Census`, `ACS-I`, `ACS-E`, as well as the image dataset `CelebA`. We provide the statistics of the datasets in Table 2. In the following, we provide a comprehensive description of each dataset used in our experiments:

- **UCI Adult**[4] [35]: This dataset is extracted from the 1994 Census database. The downstream task is to predict whether the personal income is over 50K a year. The sensitive attributes in this dataset are gender and race.

- **KDD Census**[5] [35]: This data set contains $299285$ census data extracted from the 1994 and 1995 by the U.S. Census Bureau. The data contains $41$ demographic and employment related variables. The instance weight indicates the number of people in the population that each record represents due to stratified sampling. The sensitive attributes are gender and race.

- **ACS-I**(ncome)[6] [36]: The task is to predict whether the income of an individual is above $\$50,000$. The source data was filtered to only include individuals above the age of 16, who reported usual working hours of at least 1 hour per week in the past year, and an income of at least $\$100$. The threshold of the income is $\$50,000$. We use two data from this dataset, which spans 1 year or 5 years. The sensitive attributes we consider for this dataset are gender and race.

- **ACS-E**(mployment)[7] [36]: The task is to predict whether an individual is employed and the individuals are between the ages of 16 and 90. We use two data from this dataset, which spans 1 year or 5 years. The sensitive attribute we consider for this dataset is gender.

- **CelebA**[8] [81]: The CelebFaces Attributes (`CelebA`) dataset a large-scale face attributes dataset consisting of more than 200K celebrity images and each image has $40$ face attributes. The downstream task is to predict whether the person is attractive (smiling) or not, formulated as binary classification tasks. We consider Male (gender) and Young (age) as sensitive attributes.

We also provide the statistic of the datasets as follows:

Table 2: The summary of the datasets used in our experiment.

| Dataset | Data Type | Task | Sensitive Attributes | #Instances |
|---|---|---|---|---|
| UCI Adult | Tabular | Income | Gender, Race | 48,842 |
| KDD Census | Tabular | Income | Gender, Race | 299,285 |
| ACS-Income (1 year) | Tabular | Income | Gender, Race | 265,171 |
| ACS-Income (5 years) | Tabular | Income | Gender, Race | 1,315,945 |
| ACS-Emploement (1 year) | Tabular | Employment | Gender | 136,965 |
| ACS-Emploement (5 years) | Tabular | Employment | Gender | 665,137 |
| CelebA | Image | Attractive | Gender, Age | 202,599 |

---

[4] https://archive.ics.uci.edu/ml/datasets/adult
[5] https://archive.ics.uci.edu/ml/datasets/Census-Income+(KDD)
[6] https://github.com/zykls/folktables
[7] https://github.com/zykls/folktables
[8] https://mmlab.ie.cuhk.edu.hk/projects/CelebA.html

## D.2. Baselines

We provide the details of the baseline methods employed in our experiments. We note that all baselines use fixed training (i.e., each model represents a single level of fairness), whereas our proposed YODO trains once to achieve a flexible level of fairness. The details of the baseline methods are as follows:

- **Fixed Training** [35] is an in-process technique that incorporates fairness constraints as regularization term into the objective function [23, 37]. This approach enhances the model's fairness by optimizing the regularization term during training. The regularization term is represented as $\Delta$DP, $\Delta$EO, and $\Delta$Eodd.

- **Prejudice Remover** [37] introduces the prejudice remover as a regularization term, ensuring independence between the prediction and the sensitive attribute. Prejudice Remover uses mutual information to quantify the relationship between the sensitive attribute and the prediction, thereby maintaining their independence.

- **Adversarial Debiasing** [22] involves simultaneous training of the network for the downstream tasks and an adversarial network. The adversarial network receives the classifier's output as input and is trained to differentiate between sensitive attribute groups in the output. The classifier is trained to make accurate predictions for the input data while also training the adversarial network not to identify groups based on the classifier's output.

## D.3. Neural Network Architectures.

The experiments were conducted using a Multilayer Perceptron (MLP) neural network architecture for tabular data and a ResNet-18 architecture for image data.

- **Tabular**. Tabular data is structured data with a fixed number of input features. For our experiments, we used a two-layer Multilayer Perceptron (MLP) with $256$ hidden neurons. The MLP architecture is commonly used for tabular data.

- **Image**. We used a ResNet-18 architecture for image data in our experiments. The used ResNet-18 has been pre-trained on the ImageNet dataset. And we fine-tuned it for our task. The ResNet-18 architecture is widely used for image classification tasks, as it can handle the high dimensionality of image data and learn hierarchical representations of the input features.

## D.4. Experiment setting

In our experiments, we trained the neural network using the Adam optimizer [82]. The optimization process took into account two important metrics: accuracy and fairness. To maintain focus on these metrics, we trained the neural network for fixed numbers of epochs across different datasets. Specifically, for the `UCI Adult` and `KDD Census` datasets, we trained for 2 epochs, whereas the `ACS-I` and `ACS-E` datasets required $8$ epochs of training. The `CelebA` dataset, on the other hand, demanded $30$ epochs for adequate performance. We initialized the neural networks using the Xavier initialization [83]. As for the training parameters, we set the learning rate to $0.001$. We also used different batch sizes for the training process, with $512$ being the batch size for tabular datasets and $128$ for image datasets. This difference in batch size accommodates the varying computational requirements of the datasets. Notably, we did not apply weight decay in our experiments.

# E. Experiments on Text Data

In this section, we conducted experiments on text data, specifically on comment toxicity classification using the `Jigsaw` toxic comment dataset [84]. The objective is to predict if a comment is toxic. A portion of this dataset is annotated with identity attributes like gender and race. In our experiments, we treat gender and race as sensitive attributes. Adhering to the experimental setting described in [23], we leverage BERT [85] to convert each comment into a vector and then an MLP is utilized to make predictions based on the encoded vector. This appendix presents our experiments.

## E.1. The Accuracy-fairness Trade-offs

In this section, we validate the effectiveness of achieving fairness using our proposed method YODO on text datasets. We plot the Pareto frontier [38] to evaluate both our proposed method and the baselines. The results for the `Jigsaw` dataset are presented in Figure 20. From these figures, we make the following major observation: our method can achieve a flexible accuracy-fairness trade-off during inference time. Note that the Pareto frontier of YODO is plotted using one model, while others are plotted using 20 models.

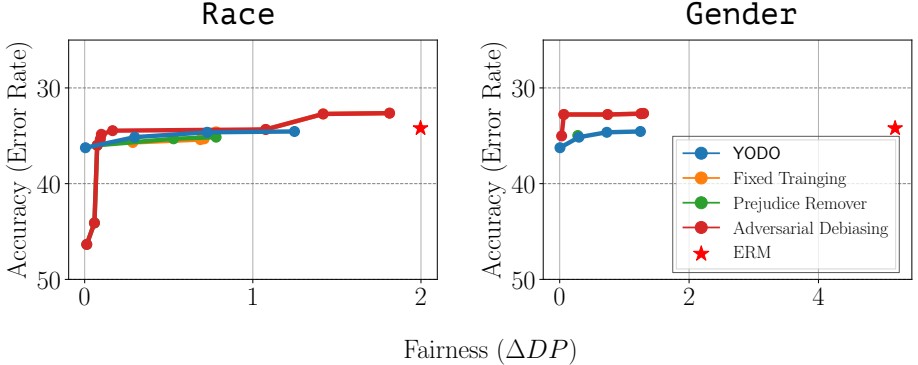

Figure 20: The Pareto frontier of model performance and fairness on the `Jigsaw` dataset based on the ΔDP metric. The sensitive attribute is gender. Note that the Pareto frontier of YODO is plotted using one model, while others are plotted using 20 models.

## E.2. Flexible Trade-offs at the Inference Time with Training Once

We explore the effect of the hyperparameter $\alpha$ on accuracy-fairness trade-offs in the `Jigsaw` datasets by varying its values during inference. The results, presented in Figure 21, indicate that $\alpha$ effectively controls the accuracy-fairness trade-offs. Specifically, as we increase the value of $\alpha$, the error rate (lower is better) rises gradually, while ΔDP diminishes.

## E.3. The Effect of Accuracy-fairness Balance Parameter $A$

In this experiment, we evaluated the performance of YODO with varying balance parameter values ($A$). We tested the model performance with varying $A$, ranging from 1 to 5, with increments of 0.5. We present the results in Figure 22 and we also display the results for each individual $A$ value in separate figures, as shown in Figure 23 and Figure 24.

The results show that YODO exhibited its best performance when $A$ is larger than 2, as this specific balance parameter value resulted in higher accuracy and a larger demographic parity span compared to other values of $A$. As the value of $A$ increases, although the fairness performance improves, the accuracy of the downstream task deteriorates. We also observed that setting $A \geq 2$ effectively addresses the trade-off between accuracy and fairness, which is slightly different than that on tabular and image datasets.

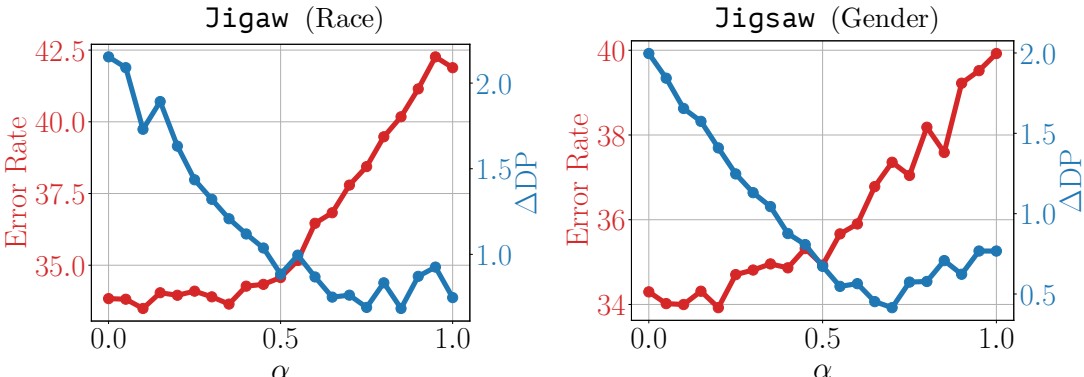

Figure 21: The accuracy-fairness trade-offs at inference time with respect to $\alpha$ on Jigsaw datasets with gender and race as the sensitive attributes. We observed that the fine-grained accuracy-fairness trade-offs could be achieved by selecting different values of $\alpha$, providing more nuanced accuracy-fairness trade-offs. Note that the results are obtained at inference time with a single trained model.

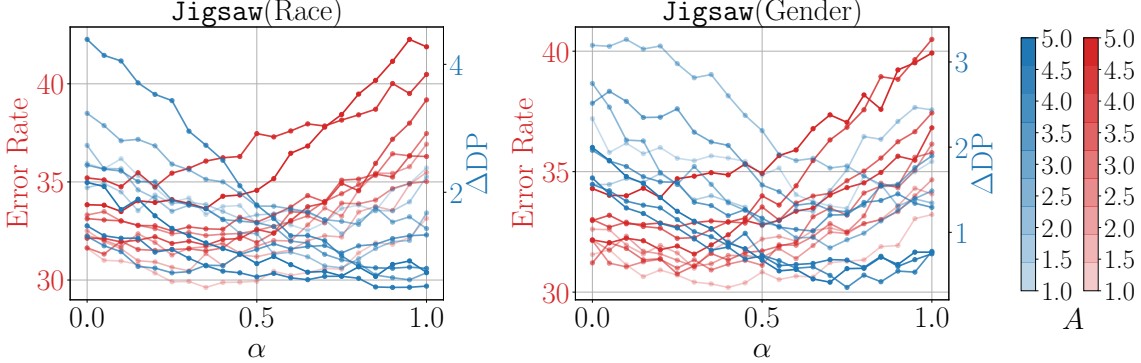

Figure 22: The effect of the accuracy-fairness balance parameter $A$ on Jigsaw dataset. The $\alpha$ in the x-axis controls the accuracy-fairness trade-off at inference time. The strength of the color reflects the value of $A$.

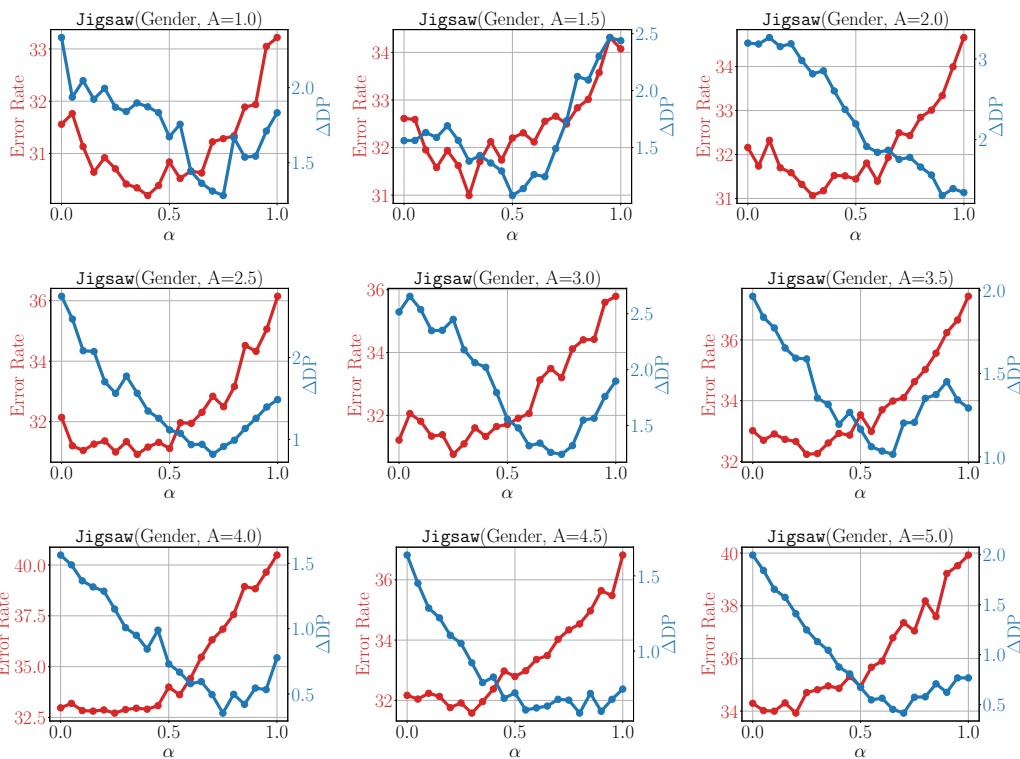

Figure 23: The effect of the accuracy-fairness balance parameter $A$ on `Jigsaw` dataset with Gender as the sensitive attribute. The results show that our method can achieve flexible accuracy-fairness trade-offs at inference time with different values of $A$.

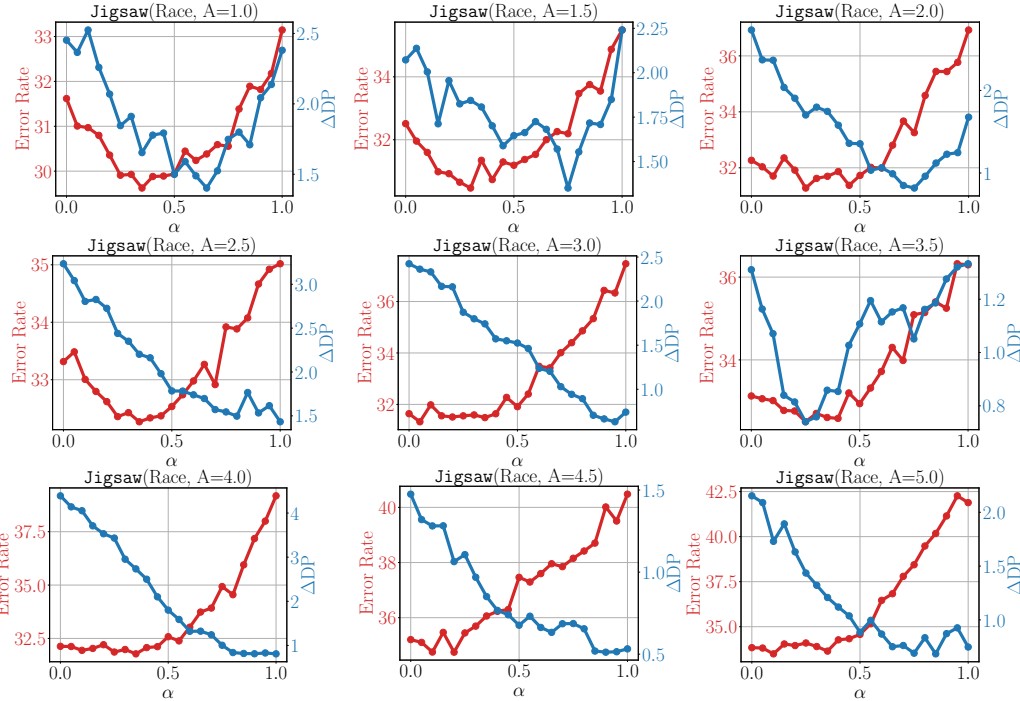

Figure 24: The effect of the accuracy-fairness balance parameter $A$ on `Jigsaw` dataset with Race as the sensitive attribute. The results show that our method can achieve flexible accuracy-fairness trade-offs at inference time with different values of $A$.

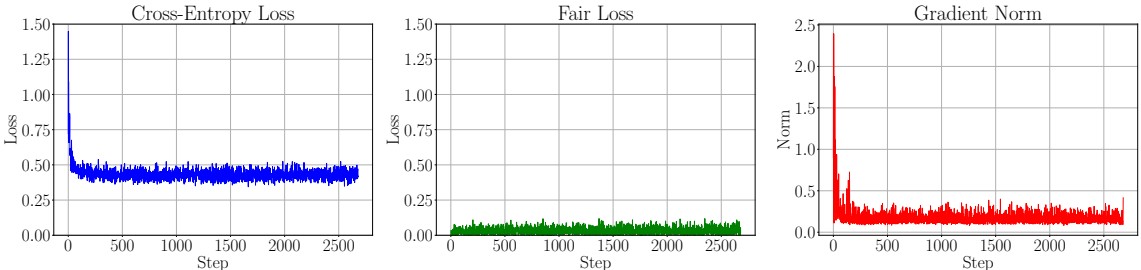

Figure 25: Training dynamics of our method. The plots show the convergence behavior during training: (a) Cross entropy loss decreases smoothly indicating stable optimization of accuracy; (b) Fairness loss remains consistently low showing maintained fairness constraints; (c) Gradient norms exhibit stable training process without major spikes or oscillations.

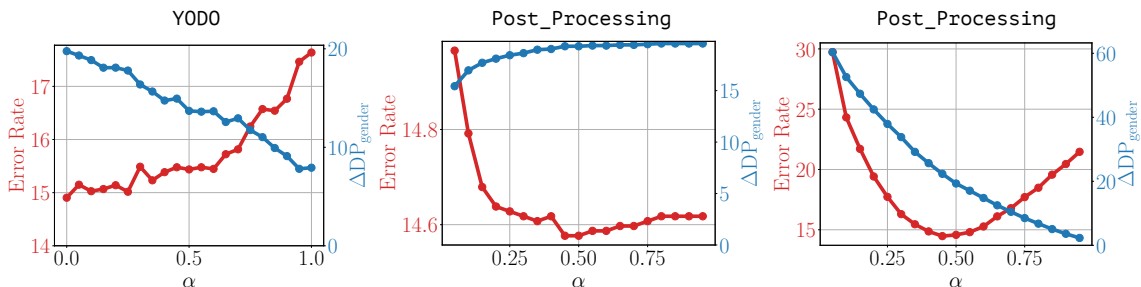

Figure 26: Comparison between our method and post-processing approaches on the accuracy-fairness trade-off. Our method achieves smoother transitions and broader coverage of the trade-off space compared to post-processing approaches.

# F.  Training Loss

To show the effectiveness of our training procedure, we plot the loss curves and gradient norms over training iterations in Figure 25. The cross entropy loss decreases smoothly and converges, indicating stable optimization of the accuracy objective. The fairness loss remains consistently low throughout training, suggesting the model successfully maintains fairness constraints. The gradient norms also exhibit stable behavior without spikes or oscillations. These results demonstrate that despite using dynamic $\alpha$ sampling, our training procedure converges reliably to an optimal balance between accuracy and fairness objectives. The smooth convergence behavior suggests that dynamic $\alpha$ sampling does not impede optimization significantly but rather enables effective exploration of the accuracy-fairness trade-off space.

# G.  YODO v.s. Post Processing

We conducted comprehensive experiments comparing our method with post-processing approaches. Specifically, we implemented threshold adjustment post-processing, where different classification thresholds are applied to different demographic groups to balance fairness. For each group, we swept the threshold from 0 to 1 in steps of 0.05 to generate different predictions. The comparison results are shown in Figure 26. The results demonstrate two key limitations of post-processing methods: (1) They cannot get a smooth trade-off curve, making it difficult to precisely control the accuracy-fairness balance. (2) The achievable curves are restricted to a limited region, whereas our method can explore the full accuracy-fairness trade-off space through continuous $\alpha$ sampling. Additionally, post-processing methods require access to sensitive attributes at test time, while our approach only needs them during training. These results highlight the advantages of our end-to-end training approach over post-hoc adjustments.

# H. Distinction Between $A$ and $\alpha$

The appendix provides a comprehensive explanation of the distinction between $A$ (in Equation (1)) and $\alpha$ (in Equation (2)).

- $A$ is a hyperparameter that controls the strength of the fairness regularization of the fairness-optimum model (✖ in Figure 1).
- $\alpha$ is a hyperparameter that controls the mixing ratio between the accuracy-optimum model (✖ in Figure 1) and the fairness-optimum model (★ in Figure 1).

Changing $A$ means changing the strongest fairness regularization of the fairness-optimum model. In Figure 7, we change the strongest fairness level (indicated by the strength level of the color of the line; for example, when $\alpha = 0$, the endpoint of the line is the fairness-optimum model and it is controlled by $A$).

Changing $\alpha$ means controlling the accuracy-fairness trade-off during inference. For example, the x-axis in Figure 5 is the $\alpha$ value; when $\alpha$ varies, the model will change to a specific point on the $\alpha$-controlled accuracy-fairness trade-off curve.)

# I. Potential Ethical Conflicts and Safeguards

A potential ethical conflict arises when using our method: if the model is deployed with two sets of weights in a fairness-regulated region, it might generate biased predictions. However, one safeguard against this is to blend the two sets of weights during deployment. By doing so, the model can produce results that adhere to specific fairness constraints. Additionally, when prioritizing fairness by deploying with two sets of weights in regions where fairness is regulated, there's a risk of compromising the model's accuracy. Yet, by judiciously combining the two sets of weights during deployment, the model can aim to meet fairness constraints while minimizing the sacrifice in accuracy.

