# OpenReview forum: "You Only Debias Once: Towards Flexible Accuracy-Fairness Trade-offs at Inference Time"
_CPAL.cc/2025/Proceedings_Track — CPAL 2025 (Proceedings Track) Oral_

### Official Review · Reviewer_9bjf · 2025-01-04
**Interesting idea, presentation could be improved**

**Rating:** 7
**Confidence:** 4

**Review:**

The submission studies how to train one model capable of inference at different points along the accuracy-fairness tradeoff curve. The proposed method uses ideas of learning a subspace of neural networks to interpolate between different points along the accuracy-fairness tradeoff curve.

Experiments on both tabular and image data are performed to study the performance of the proposed method against the baseline of training a separate model for each level of fairness desired at inference.

Strenghts:
 - The problem of inference with different levels of fairness is compelling;
 - The methodology is novel and interesting;
 - Results are promising.

Weaknesses:
 - Presentation could be improved;
 - Language is sometimes hand-wavy.

I will expand on these points and I am looking forward to discussing with the authors!

---

1. Line 86:

> which ensures that the model treats different demographic groups equally.

This sentence is imprecise. "treating different demographic groups equally" may mean different things different from demographic parity, which has a clear definition.

2. Eq. (1) and (3):

$\mathcal{L}_{f}$ is defined as a function of the model $f$ on a particular sample $x$, but $\Delta DP$ is measured over a population and is a function of $f$ only (i.e., the input is random and averaged over a population). How is the regularizer estimated during training?

I do not understand the difference between $A$ and $\alpha$. They are both multipliers of the regularizer $\mathcal{L}_f$. Later, Sec. 4.4 studies the behavior of the proposed method as a function of $A$ **and** $\alpha$. I do not understand how this comparison is carried out or what i means. Could the authors clarify the difference between $A$ and $\alpha$, and how one can change them independently, and what it means to change them? How are the curves reported in Figure 7 different from the ones in Figure 5?

3. Line 113: what does it mean to achieve $\alpha$-controlled accuracy-fairness trade-off? It does not mean that $\Delta DP$ is $\leq \alpha$. Could the authors clarify this statement?

4. Line 138: "sample a $\alpha$ during each epoch". Is a new value of $\alpha$ sampled at each epoch or at each batch?

5. Figures 3,4,9: Why does "fixed training" differ from "ERM" when $\alpha = 0$. Is this an artifact due to the variance in the training process and the resolution of the $y$-axis? If I understand correctly, "fixed training" with $\alpha = 0$ should be equivalent to ERM training. ERM seems to be consistently worse in terms of $\Delta DP$ compared to fixed training. Could the authors clarify why this happens?

It is confusing to have the y-axis grow towards the bottom of the figure. The authors could consider reporting the accuracy on the y-axis in order to improve readability of the figures.

6. Figure 5, CelebA figure: why does $\Delta DP$ grow here as a function of $\alpha$? This trend seems to be opposite to expected.

7. Lines 234-239: could the authors expand on how representations are plotted in 3D in Figure 6? What dimensionality reduction technique is used?

8. Figure 9: YODO seems to fail on KDD Census. Do the authors have any intuition behind this failure case?

Finally, it may be interested to see comparisons with existing post-processing methods.

---

**Minor comments**

- Lines 73-75:

 > The result of experiments ...

This sentence is unclear. For example, I do not understand what "comparable performance with only trained once with various neural networks" means, nor what "models for single use or fairness on a single attribute only" means. Could the authors rephrase it?

- Lines 124-125: Technically, you are not training a model with a fixed value of $\alpha$.
- Line 117: Typo in ". which".
- Line 141: "we train numerous models (infinite)". I would suggest removing the "(infinite)": everything is ultimately discretized on a machine, and it adds little to the presentation of the proposed method.
- Lines 202-203: This sentence is hand-wavy. Could the authors clarify their claim are rephrase the sentence?
- Lines 209: Could the authors clarify how smoothness of the fairness-accuracy tradeoff curve implies reliability of the model? One could have a model that performs poorly and has a smooth curve.
- Figure 7: the $x$-axis is not legible.
- Lines 260-261: Repetition in "varying prediction with the various $\alpha$.
- Line 270: Could the authors replace "higher" with "increase"?
- Lines 271-274: "explanations" is an overloaded terms that may mean very different things to diverse readers. Could the authors clarify their message in this paragraph? What are they trying to explain?
- Lines 278-279: "EO measure whether a classifier provides equal opportunities ..." is a circular definition.

---

### Official Review · Reviewer_nupQ · 2025-01-12

**Rating:** 6
**Confidence:** 3

**Review:**

Summary:

The manuscript introduces YODO (You Only Debias Once), which is a framework for achieving flexible accuracy-fairness trade-offs at inference time using a single model trained only once. The method identifies a “line” in the weight space connecting accuracy-optimum and fairness-optimum points that is able to let end-users select varying trade-off levels based on their needs. The results demonstrate that YODO achieves ultra-low overheads while offering flexible trade-offs which is able to reduce computational cost compared to training multiple models. The proposed work could be effective on applications that are improving fairness in high-stakes decision-making tasks.

Strengths:
The paper introduces a new approach for achieving flexible accuracy-fairness trade-offs during inference using a single model trained only once. This adaptability is valuable for real-world applications with diverse fairness demands instead of having the need for multiple model training.

The proposed method uses an objective-diverse neural network subspace, defined by two endpoints in the weight space (accuracy-optimum and fairness-optimum). The line between these endpoints provides transitional solutions which is to be more flexible customization of trade-offs with ultra-low computational overhead.

The proposed work shows the effectiveness and efficiency are validated across tabular and image datasets. The results demonstrate that the proposed method achieves competitive performance with lower training requirements compared to models trained separately for 1) specific fairness levels or 2) single attributes.

Weaknesses:

In Section 3, the method learns both the accuracy-optimum and the fairness-optimum endpoint in the weight space. However, I wonder if the authors could give more intuition about the reasons why the this one works instead of mentioning that the proposed work follows the [33]. What is the difference between the proposed work and [33]? Clarification is needed on learning two distinct endpoints.

I only observe that the authors use CelebA as the face attribute dataset, I wonder if the authors have tried other attribute dataset other than CelebA. There are a plenty of attribute wise dataset which could benefit the effectiveness of the propose work.

---

### Official Review · Reviewer_LKHy · 2025-01-13
**Review for Submission55**

**Rating:** 7
**Confidence:** 3

**Review:**

**Summary**:
This paper introduces YODO (You Only Debias Once), a novel method for achieving flexible accuracy-fairness trade-offs at inference time using a single model trained only once. Instead of pursuing a fixed accuracy-fairness trade-off, the approach identifies a "line" in the weight space connecting the accuracy-optimum and fairness-optimum points. This line enables users to adjust trade-offs dynamically by selecting specific positions along it during inference. The method is validated through experiments on tabular and image datasets, demonstrating its effectiveness and computational efficiency compared to traditional fairness methods.

**Strengths**:
1. **Innovative Concept**: The introduction of a dynamic accuracy-fairness trade-off mechanism at inference time addresses a critical limitation in existing fairness methods, which often require multiple models for different trade-offs.

2. **Practical Relevance**: The ability to adapt accuracy-fairness trade-offs based on varying contextual needs (e.g., legal regulations or domain requirements) makes the approach highly relevant for real-world applications.

3. **Efficiency**: The proposed method achieves flexibility with a single training process, significantly reducing computational overhead compared to training multiple models.

4. **Strong Experimental Results**: The paper provides comprehensive empirical validation on diverse datasets, showing that YODO performs comparably—or even better—than baseline methods.

5. **Detailed Analysis**: The exploration of predictive value distributions and hidden representations provides valuable insights into how the model achieves fairness.
6. **Clarity**: This paper is well-written and thoughtfully structured. It begins with a simple, illustrative 2D toy example to aid comprehension, followed by a formal definition of the core concepts.

**Weakness**:
1. **Scalability**: This paper focuses solely on binary downstream classification tasks with binary sensitive attributes. The performance of the proposed method on larger-scale datasets or more complex neural network architectures remains underexplored. This limitation could restrict its applicability to more demanding real-world scenarios where such complexities are prevalent.
2. **Simplistic Assumptions About Trade-Offs**: The assumption that all trade-offs can be effectively represented along a linear path in the weight space may oversimplify the complex relationships between accuracy and fairness. Additionally, the use of a dynamic value for $\alpha$ during the training process could impede the model's ability to converge to an optimal balance between accuracy and fairness, potentially limiting its effectiveness.
3. **Limited Discussion and Comparison with State-of-the-Art Fairness Methods**:
The paper provides only a brief discussion and limited comparison with state-of-the-art (SOTA) fairness methods. While the proposed method is innovative, a more comprehensive evaluation against existing fairness approaches, such as adversarial debiasing or post-processing techniques, would strengthen its contributions. Additionally, exploring how YODO performs relative to SOTA methods under diverse fairness metrics and scenarios could provide deeper insights into its advantages and limitations. Expanding this discussion would also help position YODO more clearly within the broader fairness research landscape.

---

### Meta-Review · Area_Chair_ciZR · 2025-02-06

**Recommendation:** Accept (Oral)
**Confidence:** 4

**Metareview:**

At a high level, this paper proposes the idea of achieving any point on the Pareto frontier between fairness and accuracy by tuning a knob for the model *at inference time*. All reviewers appreciate the novelty and potential of this contribution. The authors are recommended to closely follow the suggestions for presentation improvement.

---

### Decision · Program_Chairs · 2025-02-11

Accept (Oral)